# UNIFIED RECURRENCE MODELING FOR VIDEO ACTION ANTICIPATION

## ABSTRACT

Forecasting future events based on evidence of current conditions is an innate skill of human beings, and key for predicting the outcome of any decision making. In artificial vision for example, we would like to predict the next human action before it actually happens, without observing the future video frames associated to it. Computer vision models for action anticipation are expected to collect the subtle evidence in the preamble of the target actions. In prior studies recurrence modeling often leads to better performance, and the strong temporal inference is assumed to be a key element for reasonable prediction. To this end, we propose a unified recurrence modeling for video action anticipation by generalizing the recurrence mechanism from sequence into graph representation via message passing. The information flow in space-time can be described by the interaction between vertices and edges, and the changes of vertices for each incoming frame reflects the underlying dynamics. Our model leverages self-attention for all building blocks in the graph modeling, and we introduce different edge learning strategies that can be end-to-end optimized while updating the vertices. Our experimental results demonstrate that our modeling method is light-weight, efficient, and outperforms all previous works using the large-scale EPIC-Kitchen dataset.

## 1 INTRODUCTION

Video action recognition is a long-standing problem in computer vision. It predicts the action from experienced frames by understanding the contexts in the observations. However, having the prediction after the whole observation of action is insufficient to evaluate the outcome of any decision making. Predicting future human activities and interactions of objects are receiving research interests in these years, in assisting navigation systems (OhnBar et al., 2018), robotics (Park et al., 2016), entertainment (Liang et al., 2015; Taylor et al., 2020) and autonomous vehicles (Hirakawa et al., 2018).

Different from action recognition, the video action anticipation is required to give the predictions strictly before the action frames being observed. Models, in this case, are expected to collect the subtle evidence in the preamble of target actions (Furnari et al., 2018; Furnari & Farinella, 2020; Rodin et al., 2021). In the high-level viewpoint, the objective of anticipating is not only to understand the actual content in the observations but seeking to associate the minor details to the next possible action. Figure 1 illustrates the definition of the video anticipation problem.

In the absence of target action frames, pretrained models for the robust feature extraction from RGB inputs are often deployed on video anticipation, either pretrained from image classification or video action recognition. On top of the pretrained models, recurrent neural networks are the widely adopted options to model the temporal relationship in anticipation problems and lead to better performance than clip based modeling methods. Unlike the action recognition settings, where the relevant frames associated with the target action are not necessary in the end of observation and may locate in any action duration. video anticipation predicts the action that always performs in the future of observations, making the accumulated information gains aligned with the temporal orders. To this end, we propose a unified recurrence modeling for video action anticipation by generalizing the recurrence mechanism from a sequence into graph representation via message passing. We utilize self-attention as the universal building block in the graph modeling. In the graph point of view, self-attention can be treated as the information routing between vertices. The attention weight derived by

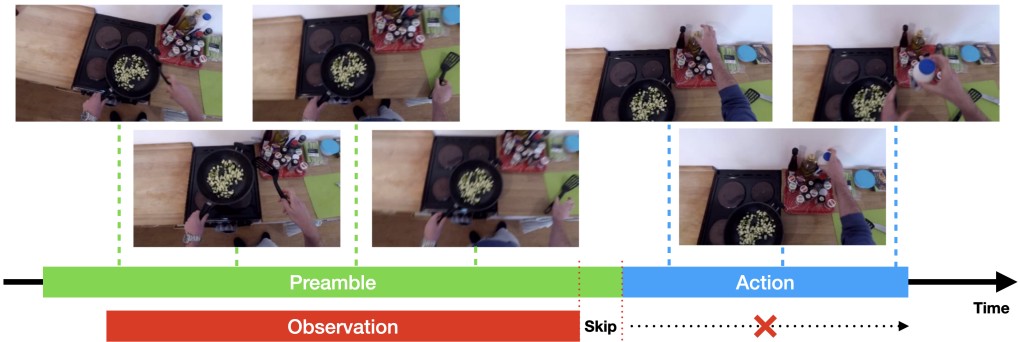

Figure 1: Illustration of video action anticipation problem. The context of action may be different than the preamble of action. Models can only observe some frames before action actually starts, which is strictly ensured by an inaccessible "Skip" period, and based on the evidence collected to predict the following action.

the scaled dot-product, which computes the correlation of vertices, can be explained as an adjacency estimation implicitly. However, in this way, the representation of edges is limited and purely based on the similarity in input tensors. Therefore, we propose three explicitly edge learning strategies trying to escape to the trivial estimation by bringing the flexibility of edge representations. They are end-to-end optimized and can be jointly trained with the main task. In the experimental results, we show our proposed unified recurrent modeling outperforms several state-of-the-art methods on large-scale egocentric video dataset EPIC-Kitchen on video anticipation. Combining with edge learning strategies, the performance can even boost to the next level.

## 2 RELATED WORKS

### 2.1 VIDEO ACTION ANTICIPATION

Early works in video anticipation model the problem with recurrent neural networks (Gao et al., 2017; Farha et al., 2018; Miech et al., 2019). Some prior works also leverage the future frames for learning the representations (Vondrick et al., 2016; Fernando & Herath, 2021). A self-regulated learning framework for action anticipation in the egocentric video is presented in (Qi et al., 2021). RU-LSTM (Furnari & Farinella, 2019) deploys two LSTMs and behaves like an encoder-decoder, where the first progressively summarizes the observed together with the second that unrolls over future predictions without observing. (Osman et al., 2021; Tai et al., 2021) both integrate unrolling mechanism as RU-LSTM does, but replace the rolling part in RU-LSTM with SlowFast (Feichtenhofer et al., 2019) and higher-order recurrent networks, respectively. (Sener et al., 2020) aggregates the predictions by pooling over different granularity of temporal segments. (Girdhar & Grauman, 2021) combine causal self-attention with the look ahead prediction on successive frames.

### 2.2 MESSAGE PASSING NEURAL NETWORK

The concept of message passing is first published in (Gilmer et al., 2017), where it was originally designed for molecular property prediction. It assumes an undirected graph structure with data-independent, equal edge contributions. To address this limitation, (Ma et al., 2020) deploys two encoders separately for vertex and edge estimation and aggregates them by a self-attention readout. Some prior works use attention or dedicated network design to learn the directed edge representations to improve the model capability (Jørgensen et al., 2018; Withnall et al., 2020; Gong & Cheng, 2019). Recently, (Arnab et al., 2021b) express the non-local attention (Wang et al., 2018) and GAT (Veličković et al., 2017) as message passing functions and apply them to video understanding task. Differently, we view the message passing framework as the generalized recurrent models and specialize it for edge representations learning in this work.

## 2.3 SELF-ATTENTION

(Vaswani et al., 2017) first propose a recurrence free sequence learning architecture by stacking several self-attention layers, which can achieve remarkable performance in the NLP domain. (Dehghani et al., 2018) demonstrates that the self-attention can be treated as the recurrent unit which unfolds to input sequences to processes with shared weights. On the other hand, (Dosovitskiy et al., 2020) propose Vision Transformer (ViT), an architecture with only self-attention for image classification. (Zhou et al., 2021) improves ViT by re-attending the multi-heads information in the post-softmax step to enable the deeper configuration. Some recent studies explore ViT based models on video action recognition (Arnab et al., 2021a; Bertasius et al., 2021), and also video anticipation (Girdhar & Grauman, 2021). Unlike these prior works, our proposed model processes the video in a flexible graph representation and fits into the message passing framework. It is lightweight and only contains few self-attention layers which sequentially process each timestep.

## 3 METHODOLOGY

### 3.1 BACKGROUNDS

#### 3.1.1 MESSAGE PASSING

Given an undirected graph $G$, the Message Passing algorithm involves two-phase forwarding processes, message passing phase with message function $M$ and update function $U$, and readout phase with readout function $R$. The message passing phase can run arbitrary $T$ step with the following definition:

$$m_v^{t+1} = \sum_{w \in N(v)} M_t(h_v^t, h_w^t, a_{vw}) \tag{1}$$

$$h_v^{t+1} = U_t(h_v^t, m_v^{t+1}) \tag{2}$$

where $v$ is the vertex in $G$, and $N(v)$ defines the neighbors of $v$. $a_{vw}$ is the connection strength bonding between vertex $v$ and $w$. The readout phase then extract the features of the whole graph, abstracted from the message passing phase, at time $T$,

$$\hat{y} = R(h_v^T | v \in G). \tag{3}$$

Our proposed method inherits these three core functions (e.g., message, update, and readout). However, the most notable difference is that we treat the whole anticipation predicting as a chain of messages conveyed in space-time, with the constant strengthening of the signal by the RGB inputs in each time-step. The readout function is called when the prediction is required at any time $t$.

#### 3.1.2 SELF-ATTENTION

Self-Attention (SA) forms the $Q, K, V$ tokens (for query, key, and value) from the same source input $x$[1]. The output of the attention is the linear combination of the value, based on the attention weight computed by the scaled dot-product between the query and key followed by a softmax.

$$Q, K, V = x W_Q^i, x W_K^i, x W_V^i \tag{4}$$

$$SA_i(x) = softmax(\frac{Q^T K}{\sqrt{D}})V, \tag{5}$$

where scaled factor $D$ default to the input feature dimension, $W_Q^i, W_K^i, W_V^i$ are the trainable embeddings. We use superscript $i$ to note the embeddings are associated to each self-attention layer.

Multi-Head Self Attention (MHSA) performs $n$-way self-attention in parallel, where $n$ is the total number of heads. An additional aggregation function, with parameters $W_{agg}$, is adopted to fuse the information computed from each head.

$$MHSA(x) = [H_1, \ldots, H_n] W_{agg}, \tag{6}$$
$$where\ H_i = SA_i(x)$$

---

[1]Some literature uses self-attention no matter the inputs $Q, K, V$ comes from the same source. We use the term, self-attention, strictly when $Q, K, V$ is formed by the same source input.

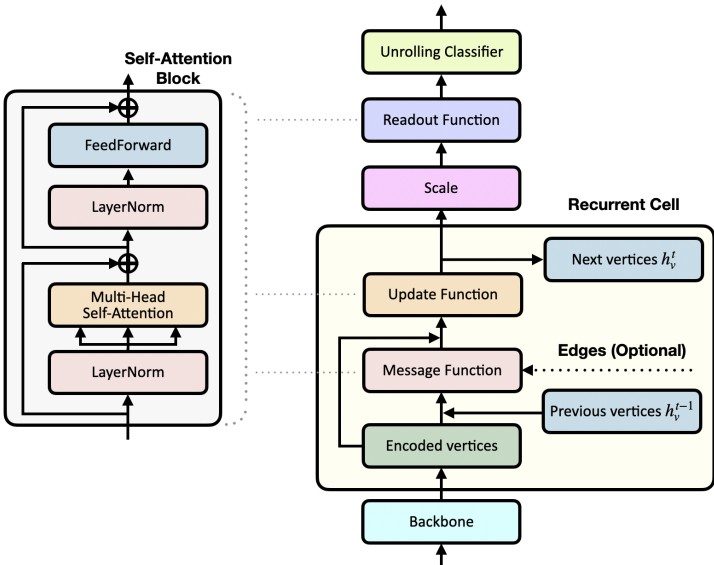

Figure 2: Overview of proposed unified recurrent model. The message function, update function, and readout function leverage multi-head self-attention. Our proposed message function is designed to be flexible to work in conjugation with explicit edges information provided.

where the $[., .]$ presents the concatenation.

Following the transformer-style architecture, a Feed-Forward Network (FFN) performs after each attention layer to project the attention output and bring the non-linearity. The FFN computes:

$$FFN(x) = \sigma(xW_1 + b_1)W_2 + b_2 \tag{7}$$

where $\sigma$ can be any arbitrary nonlinear function.

Our proposed model utilizes multi-head self-attention and deeply integrates into the message passing design. We use multi-head self-attention as the information routing between vertices of a graph. Note in this case, the resulting graph is bi-directed, because the dot-product between $Q$ and $K$ is not symmetric.

## 3.2 UNIFIED RECURRENT MODELING

**Self-Attention Block**: We implement the self-attention building block in prenorm style (Xiong et al., 2020), as implemented in many transformer-style architecture, where the normalization layer is placed before MHSA and FFN and not counts in shortcuts. Figure 2 left shows the self-attention block

$$
\begin{aligned}
f_{MHSA}(x) &= x + MHSA(LayerNorm(x)) & (8)\\
f_{FFN}(x) &= x + FFN(LayerNorm(x)) & (9)\\
SABlock(x) &= f_{FFN}(f_{MHSA}(x)) & (10)
\end{aligned}
$$

We can optionally expose the edge information, in matrix $A$, into $SABlock(.; A)$. In this case, the extension defines $MHSA(x; A)$ where the matrix $A$ is fused into the step after the softmax of scaled dot-product computation. We rewrite the equation 5 and equation 6 to

$$
\begin{aligned}
MHSA(x; A) &= [H_1, \ldots, H_n]W_{agg}, & (11)\\
where\ H_i &= SA_i(x; A)
\end{aligned}
$$

$$
SA_i(x; A) = \left(softmax(A) + softmax(\frac{(xW_Q^i)^T(xW_K^i)}{\sqrt{D}})\right)(xW_V^i) \tag{12}
$$

Note $A$ is unique and shared in multi-heads self-attention. Although the probability comes with post-softmax becomes twice larger, such difference may not cause an issue because trainable weights, $W_{agg}, W_V$, are able to absorb the impact.

**Recurrent Cell**: Figure 2 shows the overall architecture. Given the frame features $x^t$ at time $t$, with shape $(H, W, C)$, which is transformed by a backbone model, we define the vertices $e_v^t$ by taking a nonlinear transformation of $x^t$ and flatten into 1D, resulting in the shape $(HW, C)$. The position encoding $W_{pe}$ is added on top of $e_v^t$. We then leverage $SABlock(.; A)$ to message function $M$ with optionally matrix $A$, and $SABlock(.)$ to update function $U$ for computing the hidden states $h_v^t$,

$$e_v^t = f_{se}(\sigma(x^t W_x + b)) + W_{pe} \tag{13}$$

$$g_v^t = f_h([e_v^t, h_v^{t-1}]) \tag{14}$$

$$m_v^t = SABlock(g_v^t; A^t) \tag{15}$$

$$h_v^t = tanh(SABlock([e_v^t, m_v^t])) \tag{16}$$

where $f_{se}$ is a Squeeze-and-Excitation (SE) module (Hu et al., 2018) performs on vertices dimension, and $f_h$ is a linear transformer to reduce the feature size from concatenated $2C$ into $C$. Note we bound the hidden states by the $tanh$ to ensure the value stability in temporal propagation. For the readout function, we leverage $SABlock(.)$ to abstract the information from the hidden state representation $h_v^t$,

$$y^t = SABlock(f_{scale}(h_v^t)). \tag{17}$$

where $f_{scale}$ is a fully-connected transformation to scale the values of $h_v^t$. We assume every vertex in the graph $G$ is accessible by any other vertex to maintain the maximal possibility without prior. In this case, $N(v) \equiv v$ in equation 1. The edges, presents by adjacency matrix $A = \{a_{vw}; \forall w \in N(v), v \in G\}$, can be optionally provided, *explicitly*, else *implicitly* derived in self-attention.

**Implicit Edge Estimation**: The scaled dot-product original in the self-attention operator can be treated as *implicit* estimation of edges, which computes the pairwise similarity by correlation measurement of vertices on-the-fly with the parametric trainable embeddings of inputs.

**Explicit Edge Estimation**: On the other hand, edge information can also be provided by feeding a adjacency matrix $A$ during the attention computation as discussed in equation 11 and equation 12. All edge learning strategies introduced in the following all belong to *explicit* edge estimation category.

## 3.3 EDGE LEARNING

We propose three different learning strategies to construct the edge information (see Figure 3). *Edge attention* decouples the attention operator in message function into vertex and edge estimation separately. *Class token projection* performs the outer-product of a trainable vector with supervision signal from class labels, and *template bank* obtains the edge matrix by the linear combination of trainable templates, based on a selecting module conditions on inputs.

### 3.3.1 EDGE ATTENTION

We can model the adjacency matrix by deploying a dedicated $SABlock$ for edge matrix. In this case, we put two separate self-attention layers in parallel, one for vertex estimation (with subscript $v$) and another one for edge estimation (with subscript $e$). Applying to equation 15, we have

$$\widehat{A}^t = SABlock_e(g_v^t)) \tag{18}$$

$$m_v^t = SABlock_v(g_v^t; \widehat{A}^t)) \tag{19}$$

where the $\widehat{A}^t$ is the estimated edge information used in the message function.

### 3.3.2 CLASS TOKEN PROJECTION

With class token accompanied with supervision signals received, we can span an edge matrix by applying outer-product of class token. Consider a class token, $x_{cls}$, with dimension $(1, C)$, the projection can be done by a linear transformation, $f_{proj}$, from $C$ dimension to $N$. The outer-product of the token then spans it into a matrix with shape $(N, N)$, which serves as the edge matrix.

$$P = f_{proj}(x_{cls}) \tag{20}$$

$$\widehat{A}^t = P \otimes P, \tag{21}$$

where the $\otimes$ is the outer-product operator.

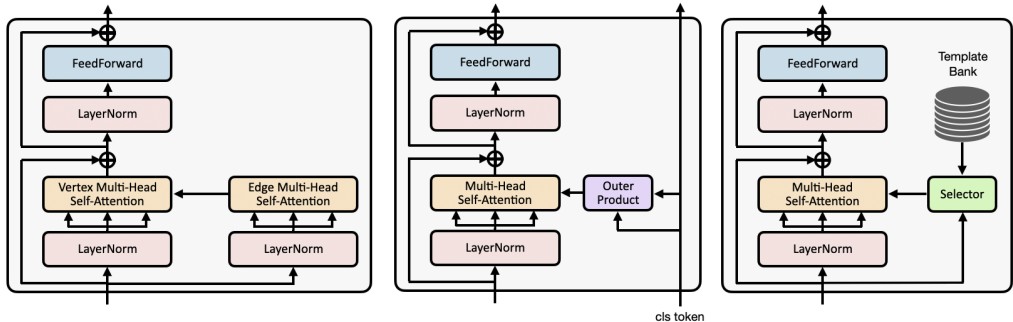

Figure 3: Propose edge learning extensions to the multi-head self-attention layer. The three block diagrams present the (left) edge attention, (middle) class token projection, and (right) template bank, respectively.

### 3.3.3 TEMPLATE BANK

We define a template bank $B$ with bank size $S$ indicates the number of templates contained. Each template in the bank is the map with the shape $(N, N)$ where $N$ is the number of vertices. During the forward process, the edge information is formed by weighted sum of $S$ templates by a selecting module, $f_{select}(.)$, a Multi-Layer Perception (MLP) followed by a sigmoid to produce the selecting ratio conditioned on inputs and apply to the bank:

$$\overline{e_v^t} = \frac{1}{N}\sum_{i=0}^{N} e_{v,i}^t \tag{22}$$

$$I = sigmoid(f_{select}(\overline{e_v^t})) \tag{23}$$

$$\widehat{A}^t = \sum_{i=0}^{S} I_{:,:,i} \cdot B_{i,:,:} \tag{24}$$

where $\overline{e_v^t}$ is the average of $e_v^t$ over the vertex dimension. The edge estimation $\widehat{A}^t$, in the combination of the template bank $B$, is then fed into the message function the equation 15.

## 4 EXPERIMENTS

### 4.1 IMPLEMENTATION DETAILS

We based the pretrained BN-Inception model of (Furnari & Farinella, 2019) as backbone and kept it frozen during training. All input frames are resized to 256x454 and fed through a proposed model followed by an unrolling classifier forming the verb, noun, and action prediction. The unrolling classifier is a widely adopted design in action anticipation (Furnari & Farinella, 2019; Osman et al., 2021; Tai et al., 2021), which unfolds the final output over the unobserved interval with a LSTM, till the moment where the action is expected to start, and then computes the predictions. The outputs are supervised by comparing predictions with ground-truth labels with cross-entropy in the last 8 anticipate interval for each individual verb, noun, and action. Note in the class token projection scheme, the class token for action is used, otherwise mean-pool over the output from readout function is performed. For training details, RandAugment (Cubuk et al., 2020) augmentation scheme is applied in all experiments. AdaBelief (Zhuang et al., 2020) in combination with look-ahead optimizer (Zhang et al., 2019) is adopted. Weight decay is set to 0.001. Learning rate is initially set to 1e-4 and cosine annealed to 1e-7 on the last 25% of epochs. We use $4 \times$ NVIDIA V100 32GB GPUs for training, with batch size set to 32 and run in total 50 epochs.

Table 1: EK55 Action Anticipation validation results using RGB at different $\tau_a$ in Top-5 action accuracy.

| METHODS | TOP-5 ACCURACY (%) AT DIFFERENT $\tau_a$ | | | | | | | |
|---|---|---|---|---|---|---|---|---|
| | 2 | 1.75 | 1.5 | 1.25 | 1.0 | 0.75 | 0.5 | 0.25 |
| DMR | - | - | - | - | 16.86 | - | - | - |
| ATSN | - | - | - | - | 16.29 | - | - | - |
| MCE | - | - | - | - | 26.11 | - | - | - |
| VN-CE | - | - | - | - | 17.31 | - | - | - |
| SVM-TOP3 | - | - | - | - | 25.42 | - | - | - |
| SVM-TOP5 | - | - | - | - | 24.46 | - | - | - |
| VNMCE+T3 | - | - | - | - | 25.95 | - | - | - |
| VNMCE+T5 | - | - | - | - | 26.01 | - | - | - |
| ED | 21.53 | 22.22 | 23.20 | 24.78 | 25.75 | 26.69 | 27.66 | 29.74 |
| FN | 23.47 | 24.07 | 24.68 | 25.66 | 26.27 | 26.87 | 27.88 | 28.96 |
| RL | 25.95 | 26.49 | 27.15 | 28.48 | 29.61 | 30.81 | 31.86 | 32.84 |
| EL | 24.68 | 25.68 | 26.41 | 27.35 | 28.56 | 30.27 | 31.50 | 33.55 |
| RU-RGB | 25.44 | 26.89 | 28.32 | 29.42 | 30.83 | 32.00 | 33.31 | 34.47 |
| HORST | 25.38 | 26.37 | 27.82 | 29.16 | 30.69 | 31.54 | 32.52 | 33.45 |
| HORST-URL | 25.95 | 27.03 | 28.24 | 29.81 | 31.58 | 32.68 | 34.21 | 35.56 |
| SRL | 25.82 | 27.21 | 28.52 | 29.81 | 31.68 | 33.11 | 34.75 | 36.89 |
| SF-RU ($\alpha_s=\frac{1}{8}$) | 24.53 | 25.63 | 27.30 | 28.97 | 30.96 | 32.23 | 33.49 | 35.02 |
| SF-RU ($\alpha_s=\frac{1}{2}$) | 26.39 | - | 28.40 | - | 30.94 | - | 32.87 | - |
| SF-RU ($\alpha_s=\frac{1}{2},\frac{1}{8}$) | 26.78 | - | 29.25 | - | 32.05 | - | 34.34 | - |
| OURS (IMPLICIT) | 27.25 | 27.76 | 29.36 | 30.63 | 31.68 | 32.76 | 34.41 | 36.65 |
| OURS (EDGE ATTENTION) | 27.25 | 27.98 | 29.24 | 30.29 | 31.52 | 32.92 | 34.77 | 36.95 |
| OURS (TEMPLATE BANK) | 26.67 | 27.76 | 29.32 | 30.49 | 32.02 | 33.47 | 34.71 | 36.85 |
| OURS (CLS PROJECTION) | 26.87 | 27.90 | 29.44 | 30.63 | 31.96 | 33.19 | 34.92 | 37.05 |

## 4.2 EPIC-KITCHEN DATASET

EPIC-Kitchen 55 (EK55) (Damen et al., 2018) is a large scale egocentric video dataset, captured by 32 subjects in 32 kitchens. The data split scheme is inherited from (Furnari & Farinella, 2019) for action anticipation tasks, resulting in 23492 action segments for training and 4979 for validation. All unique verb-noun pairs, of 125 verbs and 352 nouns. define 2513 action categories. The top-1 and top-5 accuracy at $1s$ are the major evaluation metrics used on EK55. We also include the specific performance metrics at $1s$, with additional mean top-5 recall reported. We sample 14 frames from each clip with a fixed stride of $\alpha_s = 0.25$ (4 fps), resulting in $3.5s$ context in each sample. We use $\tau_a$ to indicate anticipation time.

## 4.3 EXPERIMENTAL RESULTS

**Baselines**. We consider the baselines methods, all in RGB inputs, including Deep Multimodal Regressor (DMR), (Vondrick et al., 2016), TSN-based models MCE (Furnari et al., 2018) and ATSN (Damen et al., 2018), deep network trained with top-k classifier (SVM-Top3/5) (Berrada et al., 2018), Verb-Noun Marginal Cross Entropy (VNMCE) (Furnari et al., 2018), and several LSTM variants, Encoder-Decoder LSTM (ED) (Gao et al., 2017), Feedback Network LSTM (FN) (Geest & Tuytelaars, 2018), LSTM with Ranking Loss (RL) (Ma et al., 2016), and LSTM with Exponential Anticipation Loss (EL) (Jain et al., 2016). We also compare the state-of-the-art Rolling-Unrolling LSTM (RU) (Furnari & Farinella, 2020), ImagineRNN by predicting future feature (Wu et al., 2020), Higher-Order Recurrent Space-Time Transformer (HORST) (Tai et al., 2021), Self-Regulation Learning (SRL) (Qi et al., 2021), SlowFast with unrolling classifier (SF-RU) (Osman et al., 2021), and previous winners in EPIC-Kitchens anticipation challenge (in Table 3), Action-Banks in 2020 challenge (Sener et al., 2020) and Anticipative Video Transformer (AVT) in 2021 (Girdhar & Grauman, 2021). Some methods cannot query the accuracy at any time (i.e., DMR, ATSN, MCE, VN-CE, SVM-Top3/5, VNMCE) and some are constraint by the specific sampling rates (i.e., SF-RU, AVT). The $\alpha_s$ in SF-RU presents the data sampling rate used in the algorithm.

**Results**. From Table 1, we can observe that our proposed method with implicit edge estimation already performs strongly compared to previous methods on EK55. Using edge learning strategies discussed in section 3.3, we found template bank and class token projection methods with significant improvements. Table 2 shows the specific performance at $1s$, which is the widely used evaluation criteria in the EPIC-Kitchen, we provide the top-5 accuracy, and also top-5 recall, for each individual verb, noun, and action. We can clearly see our proposed unified recurrent modeling outperforms previous methods, not only on the accuracy but also on the recall. Equipped with the explicit edge

Table 2: EK55 Action Anticipation validation results using RGB at $1s$ in Top-5 accuracy and recall for noun, verb, and action.

| METHODS | TOP-5 ACC. (%) @ $1s$ | | | MEAN TOP-5 REC. (%) @ $1s$ | | |
|---|---|---|---|---|---|---|
| | VERB | NOUN | ACTION | VERB | NOUN | ACTION |
| DMR | 73.66 | 29.99 | 16.86 | 24.50 | 20.89 | 03.23 |
| ATSN | 77.30 | 39.93 | 16.29 | 33.08 | 32.77 | 07.06 |
| MCE | 73.35 | 38.86 | 26.11 | 34.62 | 32.59 | 06.50 |
| VN-CE | 77.67 | 39.50 | 17.31 | 34.05 | 34.50 | 07.73 |
| SVM-TOP3 | 72.70 | 28.41 | 25.42 | 41.90 | 34.69 | 05.32 |
| SVM-TOP5 | 69.17 | 36.66 | 24.46 | 40.27 | 32.69 | 05.23 |
| VNMCE+T3 | 74.05 | 39.18 | 25.95 | 40.17 | 34.15 | 05.57 |
| VNMCE+T5 | 74.07 | 39.10 | 26.01 | 41.62 | 35.49 | 05.78 |
| ED | 75.46 | 42.96 | 25.75 | 41.77 | 42.59 | 10.97 |
| FN | 74.84 | 40.87 | 26.27 | 35.30 | 37.77 | 06.64 |
| RL | 76.79 | 44.53 | 29.61 | 40.80 | 40.87 | 10.64 |
| EL | 75.66 | 43.72 | 28.56 | 38.70 | 40.32 | 08.62 |
| RU-RGB | - | - | 30.83 | - | - | - |
| HORST | 77.67 | 46.34 | 30.69 | 36.54 | 44.33 | 10.94 |
| HORST-url | 78.80 | 46.54 | 31.58 | 42.62 | 45.68 | 12.18 |
| SRL | 78.90 | 47.65 | 31.68 | 42.83 | 47.64 | 13.24 |
| SF-RU ($\alpha_s = \frac{1}{8}$) | - | - | 30.96 | - | - | - |
| SF-RU ($\alpha_s = \frac{1}{2}$) | - | - | 30.94 | - | - | - |
| SF-RU ($\alpha_s = \frac{1}{2}, \frac{1}{8}$) | - | - | 32.05 | - | - | - |
| OURS (IMPLICIT) | 78.66 | 47.93 | 31.68 | 43.67 | 47.93 | 13.19 |
| OURS (EDGE ATTENTION) | 78.54 | 47.91 | 31.52 | 44.57 | 46.87 | 12.84 |
| OURS (TEMPLATE BANK) | 78.60 | 46.86 | 32.02 | 43.63 | 46.86 | 13.58 |
| OURS (CLS PROJECTION) | 78.74 | 47.59 | 31.96 | 44.96 | 47.19 | 13.61 |

Table 3: EK55 Action Anticipation validation results using RGB with top-1 and top-5 action accuracy at $\tau_a = 1s$.

| METHOD | BACKBONE | PRETRAIN | TOP-1 (%) | TOP-5 (%) |
|---|---|---|---|---|
| RU-RGB | BNINC | IN1K | 13.1 | 30.8 |
| ACTIONBANKS | BNINC | IN1K | 12.3 | 28.5 |
| IMAGINERNN | BNINC | IN1K | 13.7 | 31.6 |
| AVT-H | BNINC | IN1K | 13.1 | 28.1 |
| AVT-H | AVT-B | IN21+1K | 12.5 | 30.1 |
| AVT-H | IRCSN152 | IG65M | 14.4 | 31.7 |
| HORST | BNINC | IN1K | 12.6 | 30.7 |
| HORST-url | BNINC | IN1K | 12.8 | 31.6 |
| OURS (IMPLICIT) | BNINC | IN1K | 13.5 | 31.7 |
| OURS (EDGE ATTENTION) | BNINC | IN1K | 13.9 | 31.5 |
| OURS (TEMPLATE BANK) | BNINC | IN1K | 13.8 | 32.0 |
| OURS (CLS PROJECTION) | BNINC | IN1K | 13.6 | 32.0 |

estimation, template bank and class token projection even shows significant improvements. Table 3 reveals top-1 accuracy at $1s$ and compared with additional candidates who won the EPIC-Kitchen official challenge in recent two years (ActionBanks and AVT). Despite the AVT trained with advanced backbone deployed, and with different sampling rate $\alpha_s = 1$ (1 fps) for $10s$ for each sample, our proposed methods are with superior accuracy in both top-1 and top-5 accuracy over all the methods using the same BN-Inception backbone. It is worth noting although the edge attention is not as strong as template bank and class token projection proposals in top-5 comparison, it gets the best amongst all top-1 accuracy.

## 4.4 ABLATION STUDY

From the previous section, we found edge learning with template bank achieved overall best according to accuracy and recall among all experiments. In this section we compare and discuss the two major hyper-parameters set in this model, the bank size and the feature dimension.

### 4.4.1 BANK SIZE

Table 4 shows the experimental results on setting different bank sizes for the template bank edge learning. We manipulate the bank size ranges from 1 to 2048. In the case of bank size being equal to 1, a unique edge matrix is shared globally across all samples and each timestep. It can be explained as placing a strong regularization to the intermediate representations before feeding into the message function, forcing the vertex constrained to the specific representation that is able to co-work with the

Table 4: Ablation on different bank sizes is set on EK55. All the numbers are in % and at $1s$ anticipate interval.

| BANK SIZE | TOP-1 ACC. | TOP-5 ACC. | | | MEAN TOP-5 REC. | | |
| | ACTION | VERB | NOUN | ACTION | VERB | NOUN | ACTION |
|---|---|---|---|---|---|---|---|
| 1 | 13.22 | 78.34 | 47.39 | 31.38 | 44.07 | 45.86 | 12.99 |
| 32 | 12.99 | 78.96 | 47.49 | 31.68 | 43.60 | 46.43 | 12.62 |
| 64 | 13.05 | 78.42 | 47.51 | 31.58 | 42.77 | 46.91 | 13.20 |
| 128 | 13.37 | 79.04 | 48.17 | 31.84 | 44.70 | 47.15 | 13.07 |
| 256 | 13.44 | 78.80 | 48.07 | 31.98 | 44.18 | 48.08 | 13.13 |
| 512 | 13.84 | 78.60 | 46.86 | 32.02 | 43.63 | 46.86 | 13.58 |
| 1024 | 13.31 | 78.38 | 48.23 | 32.08 | 42.76 | 47.75 | 13.34 |
| 2048 | 12.89 | 78.98 | 47.18 | 31.23 | 42.35 | 45.76 | 12.63 |

Table 5: Ablation on various feature dimensions set in model on EK55. All the numbers are in % and at $1s$ anticipate interval.

| FEAT. dim | TOP-1 ACC. | TOP-5 ACC. | | | MEAN TOP-5 REC. | | |
| | ACTION | VERB | NOUN | ACTION | VERB | NOUN | ACTION |
|---|---|---|---|---|---|---|---|
| HEAD DIM=32 (0.25X) | 11.14 | 78.10 | 45.25 | 28.80 | 38.76 | 43.79 | 09.28 |
| HEAD DIM=64 (0.50X) | 12.57 | 78.74 | 47.28 | 30.85 | 41.33 | 47.28 | 11.83 |
| HEAD DIM=128 (1.00X) | 13.84 | 78.60 | 46.86 | 32.02 | 43.63 | 46.86 | 13.58 |
| HEAD DIM=256 (2.00X) | 12.49 | 78.78 | 47.47 | 30.95 | 44.08 | 47.18 | 12.60 |
| HEAD DIM=512 (4.00X) | 13.31 | 78.34 | 47.73 | 30.85 | 41.87 | 48.34 | 11.87 |

global edge matrix. When the bank size is larger than 1, the constraint becomes weaker, bringing additional flexibility to the representations. We can see when the bank size is greater than 64, top-1/5 accuracy and top-5 recall are showing better results than the global template used. The peak performance is at bank size 512, and showing no improvement with more templates.

### 4.4.2 FEATURE DIMENSIONS

Table 5 shows the performance difference while varying the width of the proposed model. By default, the feature dimension is set to 1024, which is aligned with the output dimension of BN-Inception backbone. We adjust the feature dimension from $0.25x$ to $4x$. This adjustment also affects the feature dimension used in multi-head self-attentions. From the table results, we can observe the best configuration falls in the 1024 feature dimension, which is equivalent to setting each attention head to 128 feature dimensions. This ablation suggests setting the appropriate feature dimension is needed for a specific task.

### 4.5 PARAMETERS AND COMPUTATION EFFICIENCY

Our model contains only three multi-head self-attention layers with linear transformations for dimension reduction or selection. Compared to other transformer-based models, which often stack multiple self-attention layers, our modeling is lightweight and efficient. The total parameters in our model, excluding the backbone, classifier, and normalization layers, are only about $21C^2$ (21M when $C$=1024). The template bank brings about additional 6.5M parameters, including a selector with parameters $\frac{1}{4}C(S + C)$, and templates that occupy $SN^2$. In the EPIC-Kitchen dataset with $(256, 454)$ input dimension based on BN-Inception backbone, $N$ is 112. The total computation is about 40 GFLOPs per timestep.

## 5 CONCLUSION

In this work we present a unified recurrent modeling which only uses self-attention as building block to represent features through graph, and which leverages edge learning to estimate vertex semantics. On the large-scale egocentric EPIC-Kitchen dataset for the video action anticipation, we surpass the current state-of-the-art in action anticipation. Proposed model is simple, lightweight, and thus embodies the flexibility for further extensions.

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
