# OpenReview forum: "Unified Recurrence Modeling for Video Action Anticipation"
_ICLR.cc/2022/Conference — ICLR 2022 Submitted_

### Official Review · Reviewer_HJA6 · 2021-10-28

**Correctness:** 3
**Technical Novelty And Significance:** 2
**Empirical Novelty And Significance:** 2
**Recommendation:** 3
**Confidence:** 3

**Main Review:**

The paper considers a topic which has attracted interest lately. Anticipation models are known to be hard to design and optimize, and learning representations good for anticipation is not trivial, so having discussion about this topic at this conference can be of interest.
As another plus, the proposed approach is compared with several other methods, which is not that common, especially for a task which has been tackled mostly recently. This gives a good basis for other works to compare with the state of the art.

Nevertheless, I think the paper has several flaws would be ideal to address in order to make it suitable for publication in this venue.

1) The main novelty of the paper seems to be the proposal of a method for action anticipation based on a graph representation and message passing. While these kind of approaches have not been much investigated for this problem, there is at least one work which the authors should compare with in order to put the scope of their work in the right context [A]. Apart from this missed comparison, the paper lacks an in-depth discussion on the main relation between the proposed work and the current art. In particular, it would be interesting to answer the following questions: In which ways the proposed approach is different from the current ones? What are the main advantages of the proposed approach? I would suggest revising the paper to make these points more explicit and put the proposed work in the right context, by explaining how it advances our current understanding of the problem at hand.

 2) Another major flaw of the paper is its lack of clarity in several points. I found the description of the method often hard to follow, partially due to the presence of unclear statements. Some of these statements are hard to grasp, while others reduce readability. I would suggest revising and proof readning. Some examples are reported in the following:
 - Page 1: "making the accumulated information gains aligned with the temporal orders" -> what is it meant by this statement? Is there a proof in the paper that this actually happens? What is it meant by aligned? How can it be measured?
 - Page 4: "The message function, update function, and redout funciton are leveraged multi-head self-attention" -> please revise (maybe "leverage multi-head self-attention"?)
 - Page 5: "Given the frame features $x^t$ at time $t$, with shape $(H,W,C)$, which is transformed by a backbone model." -> this sentence is incomplete, please revise
 - Page 5: "In this case where $N(v) \equiv v$ in equation 1." -> this sentence is also incomplete.
 - Page 5: "All edge learning strategies introduced following are all belong to explicit edge estimation category" -> please revise (maybe "introduced in the following all belong...")

3) The paper reports experiments on EPIC-KITCHENS-55, while an extension to the dataset, EPIC-KITCHENS-100 [B] has been introduced. Previous works also reported results on EGTEA-GAZE+ [C]. I think it is hard to draw general conclusions from experimenting on a single dataset, and it would have been interesting seeing experiments on these other datasets as well for completeness.

4) While experiments include comparisons with several other approaches. All results are restricted to the case in which only RGB inputs are processed. This is odd, as previous works have been demonstrated to be able to leverage multimodal information. For instance, RULSTM and Action Banks both use RGB, Flow and object inputs. I would have expected the paper to report comparisons with methods taking these inputs as well (I believe these would outperform the results currently reported in the paper). Additionally, it would have been interesting to study whether the proposed approach can work well also on the other representations apart from RGB. This would be natural to assess, as also Flow features can be obtained with a 2D CNN backbone and object features are proposed in RULSTM as 1D vectors, which are inputs compatible with the proposed approach.

[A] E. Dessalene, C. Devaraj, M. Maynord, C. Fermuller and Y. Aloimonos, "Forecasting Action through Contact Representations from First Person Video," in IEEE Transactions on Pattern Analysis and Machine Intelligence, doi: 10.1109/TPAMI.2021.3055233.

[B] Damen, Doughty, Farinella, Furnari, Kazakos, Moltisanti, Munro, Price, Wray (2021). Rescaling Egocentric Vision . International Journal on Computer Vision (IJCV), abs/2006.13256 .

[C] Li, Yin, Miao Liu, and James M. Rehg. "In the eye of beholder: Joint learning of gaze and actions in first person video." Proceedings of the European Conference on Computer Vision (ECCV). 2018.

**Summary Of The Paper:**

The paper proposes a method for action anticipation based on a graph representation of the temporal structure of video. Specifically a graph is built using semantic representations of video frames extracted with a 2D CNN backbone as vertices. Temporal reasoning is hence achieved modeling spatiotemporal information flow thorugh message passing across vertices. The approach is based on a message function, an unpdate function and a readout function, all based o a multi-head self-attention block. Experiments compare the proposed approach with respect to other approaches suing RGB inputs on EPIC-KITCHENS-55.

**Summary Of The Review:**

All in all I think the paper tackles an interesting problem, but its presentation lacks clarity, which makes the paper hard to follow at times and difficult to compare to current art. Also, the experimental validation is lacking in some aspects, which should be fixed to make the paper a solid contribution.

---

> ### Author Response · Authors · 2021-11-23
> **Response to reviewer HJA6**
>
> We thank the reviewer for insightful comments and point out some flaws in the writings, we have revised our manuscript according to those suggestions.
>
> ("making the accumulated information gains aligned with the temporal orders") Since the predicted action always happens later than anticipated intervals, we assume the received information gains while approaching to where the action starts.  Although this is not strictly proved by any work by others, the evidence can be found in table 1 that when the anticipated interval decreases, the performance increases.
>
> (other datasets) We appreciate the reviewer suggesting to also compare with other datasets. Although we cannot provide additional comparison numbers in the rebuttal window in time, we will definitely take these constructive suggestions as our future work.
>
> (RGB modality) Although we conducted experiments on RGB only modality, it is expected that our model can also be applied to the different modality as well, like from optical flow or features from detected objects. However, there exists more than a single way to fuse the information from multimodality, and the discussion of multimodality fusion is beyond the scope of our study. In order to have each method compared on a fair basis, we only consider RGB-only modality in this work to present and compare the effectiveness of our unified recurrent design with baselines.

---

### Official Review · Reviewer_iyEe · 2021-10-30

**Correctness:** 3
**Technical Novelty And Significance:** 3
**Empirical Novelty And Significance:** 3
**Recommendation:** 3
**Confidence:** 4

**Main Review:**

Strength:

+The message passing method for learning graph representation under the video anticipation setting is novel to my knowledge

+The idea of using self-attention learning the correlation between vertices and further building the graph representation is interesting.

+Different edge learning methods are proposed and properly evaluated in the ablation study

+The experiment results seem to be promising

+The paper is overall well-written


Weakness (sorted by priority):

-The major concern is the effectiveness of the proposed three edge learning strategies. According to Table1-3, those 3 strategies have minor improvement on the overall performance.

-The method section is not well-structured. It will be nice to have an overview of how to build the graph of the video sequence with proposed operations

-It would be great to compare with previous methods on the efficiency.

-Some visualization of what has been learned by the self-attention would be interesting, and may potentially improve the paper quality/

-Only results on EPIC-Kitchens are shown. Many baseline methods also conduct experiments on the EGTEA Gaze+ dataset.

-missing reference:

EGO-TOPO: Environment Affordances from Egocentric Video CVPR 2020

Forecasting Human Object Interaction: Joint Prediction of Motor Attention and Actions in First Person Vision. ECCV2020



**Summary Of The Paper:**

This paper addressed an interesting problem of action anticipation. To this end, the authors proposed a unified recurrence model that generates the graph representation of the video sequence. Extensive experimental results have shown the effectiveness of the proposed method.

**Summary Of The Review:**

The overall idea of using message passing for constructing graph representation for video action anticipation. But the proposed edge learning method has a minor influence on the model performance. Therefore some of the arguments made by the paper are not fully supported.

---

> ### Author Response · Authors · 2021-11-23
> **Response to reviewer iyEe**
>
> We thank the reviewer for their comments and suggestions.
>
> (effectiveness of the proposed three edge learning strategies)  The self-attention assumes the vertices are fully-connected and derived via the scaled dot-product. Our explicit edge learning strategies leverage the auxiliary matrix with the scaled dot-product to complement the attention map. We observe some improvements achieved by edge learning, although not major, but in a reasonable range.
>
> (graph representation) We treat each of the corresponding pixels after the backbone extractor as the vertices of the graph, and update the vertices across the temporal dimension via a message passing framework.
>
> (efficiency) We analyze the time and space complexity of our proposed method in section 4.5. Since most of the comparison baselines didn’t provide the efficiency analysis, we didn’t put the head-to-head efficiency comparison on the result table.
>
> (Visualizations, more comparisons, and missing references) We thank the reviewer for providing many constructive suggestions. Although we cannot finish the additional experiments in this rebuttal window, we will indeed consider those directions for our work in future.

---

> > ### Comment · Reviewer_iyEe · 2021-11-29
> > **Clraification fo Methodology & Experimental Results**
> >
> > The initial concerns on the effectiveness of edge learning strategies have not been addressed during the rebuttal. The claimed "reasonable" range is vague and can not support the model design.
> >
> > Some details are missing in the methodology section.
> >
> > Recommendation: Though this paper has certain merits, it is not ready for publication in its current form. Therefore, I do not recommend acceptance for this paper. I encourage the authors to further polish the paper and add the experimental results.

---

### Official Review · Reviewer_hNXw · 2021-11-01

**Correctness:** 4
**Technical Novelty And Significance:** 2
**Empirical Novelty And Significance:** 3
**Recommendation:** 6
**Confidence:** 3

**Details Of Ethics Concerns:**

No concerns.

**Main Review:**

Strengths
1. Novel approach which solves an important vision task: video action anticipation.
2. Method is clearly and methodically described.
3. Experimental results are strong on the challenging EK55 dataset. The approach is compared against many state-of-the-arts algorithms.

Weaknesses
1. Paper's approach is focus on the specific task of video action anticipation task. While it is an important task in vision, it is not clear if it is of interests to the wider audience in the ICLR community.
2. Minor. Paper's main technical contributions are specific to the task and may not generalise to other problems.

**Summary Of The Paper:**

Paper proposes a novel architecture for video action anticipation task. The proposed method used a graph representation via message passing. The key contribution lies in the three explicitly edge learning strategies. These strategies to escape to the trivial estimation which is  purely based on the similarity in input tensors, by bringing the flexibility of edge representations.

1. Edge attention decouples the attention operator in message function into vertex and edge estimation separately.
2. Class token projection performs the outer-product of a trainable vector with supervision signal from class labels.
3. Template bank obtains the edge matrix by the linear combination of trainable templates, based on a selecting module conditions on inputs.

Experiment on the EK55 dataset outperforms the state-of-the-arts algorithms.

**Summary Of The Review:**

I recommend to accept this paper as the technical contributions are quite strong. However, its appeal to the wider audience in ICLR is a minor concern.

---

> ### Author Response · Authors · 2021-11-23
> **Response to reviewer hNXw**
>
> We thank the reviewer’s positive comments. We leverage message passing framework, which is a general recurrent modeling, and seeing whether it can perform well in the context where the recurrent networks play an important role (i.e., action anticipation). The self-attention is the main building block in our modeling, which is also without any task specific design. We assume our proposed model can be applied and generalized to other task as well (i.e., early recognition) and left for future work.

---

### Official Review · Reviewer_yqiW · 2021-11-01

**Correctness:** 2
**Technical Novelty And Significance:** 3
**Empirical Novelty And Significance:** 2
**Recommendation:** 5
**Confidence:** 5

**Main Review:**

Strengths:

a) The proposed approach provides a transformer-based method to encode Spatio-temporal features from a graph or RGB inputs. Adding the edge affinity matrix to the attention map allows the model to make use of edge affinity information in a self-attention encoder block.

b) The task of video action anticipation can arguably aid in learning more robust and transferable representations of inputs by forcing the model to predict further away actions.

c) Unlike other methods (Furnari et al.), this method does not require optical flow or object detections as inputs
The paper is well written, and claims are supported by empirical results (Tables 1,2, and 3).

Weaknesses:

a) The authors present the architecture using GNN terminology (message, update, readout functions). Generally, GNNs are useful in the following scenarios where:

1) The input graphs are not fully connected, so the adjacency (or affinity) matrix introduces an inductive bias on the data forcing the nodes to pass messages in a certain manner, and inhibit message passing for disconnected nodes.

2) The input graphs are fully connected, but affinity matrix (or edge embeddings) influence the message passing function.

3) Permutation invariance is needed.

In the proposed approach, the authors use a transformer architecture to process the image patches. The vertices (image patches) of the graph are flattened into a 1D transformer sequence. These vertices are updated with every new observation using a recurrent model (similar to LSTM) where the input, update, and output functions (gates) are implemented with self-attention blocks. The input data contains no edge information between image patches. And positional encoding is added to the input, allowing the network to be aware of the positional information of patches. It becomes hard to convince the readers that GNNs terminology and edge learning is necessary unless it is well supported by empirical results. However, the results of explicit edge learning do not outperform implicit edge learning through attention, because the attention operation is capable of capturing the spatial dependencies between image patches (as implied by the authors). Introducing the edge affinity matrix into the attention block will only be useful if it is part of the input; it can be used as a prior to influence the attention maps. This framework can be more useful for different tasks with edges defined as part of the input.

b) More empirical results can be provided to discuss Tail classes (similar to table 3 in AVT paper) and more results on EK100 or a different dataset. Providing results on multiple datasets helps prove the efficacy of the approach in different settings (egocentric vs. third-person, indoor vs. outdoor, etc.).

c) The claim that edge learning with template bank achieved the best results is not well supported by quantitative results. Ablations on the main architecture (instead of template bank size) would be more useful, i.e. different backbones, removing squeeze and excitation, etc. Ideally, this can be fixed with an additional table showing the Top-1/Top-5 performance with different backbone variations and initializations. This will allow the reader to understand which part of the main architecture causes the model to perform better than the previous approaches.

d) The authors claim to surpass current SOTA approaches but based on the reported results, improvements are not significant (and not consistent).


Minor Issues:

a) Please provide an explanation for how the output of Eqn. 18 can be inserted in Eqn. 19. The output of SABlock is an (Nxd) tensor and we need an NxN attention matrix in SABlock_v. One option is to extract the attention from SABlock_e before multiplying by the Value matrix but it is not clear in the text.

b) It would be a good idea to create a separate section for the decoder (unrolling classifier) and the loss function instead of briefly mentioning them in the implementation details. The background section can be shortened to save space.

c) Some implementation details are missing, such as:
1) Is the backbone frozen or just pretrained?
2) How many blocks of SA are used? just one for every function?
3) What is the number of patches (HW) used as input to the transformer?
4) What initialization is used for node states, template bank, etc.?
5) Any data augmentation applied to the dataset? horizontal flipping seems like a good fit for this data.
6) What is the function f_{scale} scaling the values of h_{v}^{t} to?
7) Minor typos are present (i.e., Section 4.1:L2)


Ideas that are not required for publication, but would make the paper stronger:

a) It would be an interesting idea to test the learned features on the action recognition task. Using this architecture as a pretrained encoder for an action recognition task may outperform other approaches with naive initializations. The would be a very strong section in the evaluation.

b) Other papers (Furnari et al.) use object detections and optical flow features as part of the input. It would be interesting to see the effect of including these features as input to the model.


**Summary Of The Paper:**

The authors propose a framework for video action anticipation, where the task is to observe input frames and predict the action label after an anticipation period (not observed). The proposed architecture builds on top of message passing terminology (message, update, and readout functions) to create a recurrent transformer model that uses self-attention to capture spatial dependencies between frame patches (similar to ViT w/o [cls]) and recurrent module to capture the temporal dependencies across frames. Edge affinity matrix is learned and introduced into the MHSA block (Eqn. 12) as a prior to influence the attention matrix at every layer. Three different explicit edge learning approaches are proposed, in addition to implicit edge learning. The model is evaluated on the Epic Kitchen 55 dataset.

**Summary Of The Review:**

The authors propose an interesting architecture for encoding RGB inputs and predicting actions after the anticipation period. The method has the capacity to encode edge information of graphs in the transformer model, but the type of data/task does not make use of this advantage. Other approaches (AVT) provide very similar solutions by using a temporal transformer, instead of a recurrent model, to capture the temporal dependencies. The GNN analogy seems weak and distracts the reader from the main approach (recurrent transformer). It can be fixed by simply removing the GNN and edge related sections, this will save space for additional ablations and quantitative results on other datasets.

The claim that this approach surpasses the SOTA approaches is weakly supported by quantitative results. The slight overall increase in performance cannot be attributed to explicit edge learning (as can be seen in the results). It is unclear which component(s) of the proposed architecture is the cause; it could be the added squeeze and excitation block, or bypassing the bottleneck representation (by not using a [CLS] token). The paper is not ready for publication in its current state; however, additional ablation studies, experiments on additional datasets, and removing unnecessary sections can significantly increase the paper's chances for publication.

---

> ### Author Response · Authors · 2021-11-23
> **Response to reviewer yqiW**
>
> We thank the reviewer for their comments and suggestions.
>
> (edge learning) We agree with the reviewer that the edge information can be more useful if it is defined by the additional priors. However, In most cases such prior is not trivial to obtain, especially in vision problems. In this work we therefore try to complement the attention mapping by feeding with the affinity matrix derived from explicit edge learning, which provides different learned features to the scaled dot-product used in self-attention.
>
> (experiments and ablations) We thank and agree with the reviewer that more experiments or ablations can help. Unfortunately, we cannot finish the additional experiments in time during this rebuttal window. However, we will take these constructive suggestions as the future work to improve our paper.
>
> (unrolling classifier) we add the details about the unrolling classifier in the supplementary.
>
> (how the output of Eqn. 18 can be inserted in Eqn. 19.) As we described in section 3.2 and Eqn. 12., the edge matrix is with shape NxN and adds to the attention weighting, also NxN, before multiplying on value.
>
> (Some implementation details are missing)
> (1) Backbone is frozen all the time.
> (2) We only deployed a single SABlock for each message passing function. Considering the model is recurrent, the effective depth of the model is sufficient.
> (3) Since backbone already reduces the inputs to the size 8x14, we take each corresponding pixel of the extracted feature map as the token (i.e.,  patch).
> (4) We didn’t particularly use customized initialization in our network. All the weights and parameters are initialized with PyTorch default initialization.
> (5) As described in section 4.1 we utilize RandAugment
> (6) Considering that $h_{v}^{t}$ is bounded in [-1, 1] for recurrent stability by hyperbolic tangent, we use a linear transformation on $h_{v}^{t}$ to loosen this constraint before readout function.
> (7) We thank the reviewer for pointing out the typo in section 4.1.

---

> > ### Comment · Reviewer_yqiW · 2021-11-29
> > **Missing implementation details have been provided. More experimentation on different datasets, and investigation into the explicit learning technique is required before publication.**
> >
> > (edge learning) I agree that such priors (relations between image patches) are hard to obtain. It may be beneficial to explore relations between objects and concepts as a prior (conceptNet is an example). The use of different learned features can be interpreted as a variant of ensemble methods. If this is your intention, it would be good to mention it in the paper.
> >
> > (Eqn 18) My understanding is that the SABlock outputs a sequence of the same size as its input (after multiplying by value and after the FFN block). It is slightly confusing that the output of a SABlock is an A matrix in eqn.18. This can be solved by possibly renaming the SABlock to something else.
> >
> > Recommendation: I still believe that the paper (in its current state) is not ready for publication.
> > The idea of a recurrent model on top of a vision transformer is interesting, but more experimentation is needed to support the authors' claims.
> > The provided results do not support the claim that explicit edge learning allows the model to learn better relations between the image patches.

---

### Author Response · Authors · 2021-11-23
**We are greatly thankful for all reviewer’s insightful comments**

We are greatly thankful for all reviewer’s insightful comments and felt interested in our proposed idea. We fully agree more experiments will be helpful, unfortunately this cannot be done in time during this rebuttal window and will be left as our future work. We address technical concerns raised from reviewers and reply individually below.

---

### Decision · Program_Chairs · 2022-01-20

**Decision:**

Reject

**Comment:**

This paper presents work on action anticipation.  The reviewers appreciated the message passing based method.  However, concerns were raised regarding novelty, effectiveness, presentation, empirical results, and magnitude of impact for ICLR.  The reviewers considered the authors' response in their subsequent discussions but felt the concerns were not adequately addressed.  Based on this feedback the paper is not yet ready for publication in ICLR.